# Integrated Analysis of an Innovative Composite Polycaprolactone Membrane and a Jason Membrane in Guided Bone Regeneration

**DOI:** 10.3390/bioengineering13010023

**Published:** 2025-12-26

**Authors:** Alexandra Papuc, Simion Bran, Marioara Moldovan, Gabriel Armencea, Bogdan Crisan, Liana Crisan, Grigore Baciut, Cristian Dinu, Florin Onișor, Winfried Kretschmer, Mihaela Baciut

**Affiliations:** 1Department of Maxillofacial Surgery and Implantology, Iuliu Hațieganu University of Medicine and Pharmacy, Iuliu Hossu Str. 37, 400029 Cluj-Napoca, Romania; dr.alemuresan@gmail.com (A.P.); dr_brans@umfcluj.ro (S.B.); bbcrisan@yahoo.com (B.C.); petrutliana@yahoo.com (L.C.); compartiment.maxilo@scjucluj.ro (G.B.); cristian.dinu@umfcluj.ro (C.D.); florin.onisor@umfcluj.ro (F.O.); mbaciut@umfcluj.ro (M.B.); 2Raluca Ripan Institute for Research in Chemistry, Babeș Bolyai University, Fantanele 30, 400294 Cluj-Napoca, Romania; iccrr@ubbcluj.ro; 3Klinik fur Mund-, Kiefer- und Plastische Gesichtschirurgie, Alb Fils Kliniken GmbH, 73033 Goppingen, Germany; winfried.kretschmer@af-k.de

**Keywords:** resorbable membrane, polycaprolactone material, polycaprolactone vs. Jason

## Abstract

In the context of guided bone regeneration (GBR), the selection of an appropriate resorbable membrane plays a crucial role in the clinical success of the procedure. Precise knowledge about the distinct differences in properties is fundamental for correct selection of the membrane. This article presents an integrated comparative analysis between membranes, conducted for this given purpose and one step beyond: to fabricate a novel membrane with dedicated enhanced properties according to the targeted function. Our previous analysis showed that polymer membranes that met most histopathological criteria also produced the most remarkable results when radiologically observed. The most effective scaffolds were those containing active macromolecules released conditionally and staged. The PLGA and polycaprolactone scaffolds were found in this category and they granted a marked increase in bone density and improvement in osteoinduction. Based on these results, we decided to create a new polycaprolactone membrane in order to compare it with a standard currently on the market, the Jason membrane. The Jason^®^ membrane is a natural collagen scaffold derived from porcine pericardium. Due to the unique production process, the membrane shows a natural honeycomb-like, multilayered collagen structure with an increased content of collagen type III, leading to remarkable tear resistance and a slow degradation rate. Also, the low thickness of 0.05–0.35 mm facilitates the soft tissue management. The Jason scaffold was compared to an innovative synthetic membrane based on polycaprolactone (PCL), focusing on their physicochemical characteristics, biological behavior, and clinical applicability. The Jason^®^ membrane was distinguished by its high biocompatibility and rapid integration, while PCL offered superior mechanical stability and long-term durability, making it a preferred option for complex or customized 3D regenerations. Based on this integrated analysis, we fabricated an innovative electrospun PCL membrane, enriched with a novel synthesized nanohydroxyapatite, in order to enhance its specific properties for the beneficial use in targeted reconstructions.

## 1. Introduction

Bone regeneration is a complex and tightly orchestrated sequence of physiological events that occurs continuously throughout life, as bone tissue undergoes constant repair and remodeling. Conventional reconstruction approaches, such as autografts and allografts, present significant limitations in addressing complex defects and restoring the native structural and functional architecture. In this context, tissue engineering offers a promising therapeutic strategy focused on regenerating rather than merely replacing damaged tissue [1].

Guided bone regeneration (GBR) relies on the use of resorbable or non-resorbable barriers to exclude soft tissues from the bone regeneration site. Within bone tissue engineering (BTE), biomaterials serve as artificial extracellular matrices that support cell adhesion, proliferation, and differentiation, thereby promoting the formation of newly developed bone tissue [2,3]. The complex interplay among proteins, proteoglycans, angiogenic growth factors, and matrix-associated vesicles within the extracellular matrix (ECM) directs host cells toward regenerative behaviors, promoting tissue-specific differentiation and injury resolution [4]. However, advanced fabrication of ECM-based materials remains challenging, and current clinical applications of ECM scaffolds are largely confined to native tissue forms or processed powders. A broad range of collagen-based scaffolds has been developed for various regenerative applications, with these structures being designed to elicit favorable biological responses and to mimic the properties of native extracellular matrices [5].

This work provides a critical assessment of the hierarchical structure and key properties of native collagen, with particular reference to the Jason MegaGen membrane [6], and examines the current challenges associated with the fabrication of polycaprolactone-based materials. Specific attention is given to the need for tailoring their physicochemical and mechanical properties, as well as their geometrical configurations, to meet the requirements of targeted biomedical applications, with a special emphasis on advanced solutions for bone tissue engineering [7,8,9]. Jason^®^ Membrane (MegaGen) is widely utilized due to their rapid biological integration; however, in recent years, synthetic membranes made of polycaprolactone (PCL) have gained popularity due to their remarkable physicomechanical properties [1,10].

Research indicates that PCL membranes not only provide superior mechanical strength but also enhance cellular proliferation, contributing to more effective bone regeneration outcomes (Table 1). The ongoing advancements in GBR materials underscore the necessity for continued investigation in this field to optimize clinical results and improve patient quality of life. The key characteristics compared between the Jason^®^ membrane and synthetic membranes, such as those made from polycaprolactone (PCL), in the context of guided bone regeneration (GBR) include [10,11,12,13,14,15,16]:

These characteristics (Table 2, Table 3 and Table 4) help clinicians decide which membrane may be more suitable for specific GBR cases, depending on the desired outcomes and clinical requirements. The selection of membrane type in guided bone regeneration (GBR) procedures should be tailored to the defect morphology. For horizontal defects, collagen-based membranes such as Jason^®^ are generally preferred due to their rapid tissue integration, high biocompatibility, and faster resorption, as these defects typically require less long-term mechanical support. In contrast, vertical defects, which demand prolonged maintenance of regenerative space and higher resistance to soft tissue pressure, are better suited for polycaprolactone (PCL)-based membranes. PCL membranes provide superior mechanical stability and slower resorption, ensuring sustained support throughout the regeneration process [17,18].

The objective and innovative aspect of this research involved the synthesis of novel nanohydroxyapatite (nHAP) using two different dispersing agents, as well as the application of this new nHAP in developing a new electrospun membrane (EM) based on polycaprolactone (PCL) [10]. The newly synthesized nHAP was characterized using transmission electron microscopy (TEM), X-ray diffraction (XRD), and attenuated total reflectance Fourier transform infrared spectroscopy (FTIR-ATR). Additionally, all precursors and electrospun membranes were structurally characterized through XRD and FTIR-ATR analyses. For mechanical characterization, parameters such as maximum load force, Young’s modulus, stiffness, and tensile strength of the electrospun membranes were evaluated. Scanning electron microscopy (SEM) was employed to investigate the topography of the membranes.

## 2. Materials and Methods

### 2.1. Materials

Polycaprolactone (PCL) with a molecular weight (M.W.) of 80,000 g·mol^−1^, gentamicin sulfate (GEN), chloroform, methanol, poly(vinyl alcohol), diammonium hydrogen phosphate ((${NH}_{4}$)2${HPO}_{4}$), calcium nitrate tetrahydrate (Ca(NO_3_)_2_·4H_2_O), and ammonium hydroxide (NH_4_OH) solution were purchased from Sigma-Aldrich GmbH, Steinheim, Germany. Darvan 821A was obtained from R. T. Vanderbilt, Norwalk, CT, USA. Darvan 821A is an ammonium polyacrylate primarily used as a dispersing agent to prevent the formation of aggregates of nanohydroxyapatite (nHAP) particles in homogeneous and highly concentrated colloidal suspensions of nHAP. Deionized water was used in all experiments. The water hardness was 18.2 MΩ·cm at 25 °C, corresponding to an electrical conductivity of 0.055 µS/cm. All commercially available materials were used without further purification.

### 2.2. Methods

#### 2.2.1. Synthesis of nHAP

Nanohydroxyapatite (nHAP) was synthesized using a wet chemical method with calcium nitrate and ammonium hydrogen phosphate as calcium and phosphorus sources. Two solutions, each containing 1200 mL, were prepared in separate 3000 mL beakers by mixing calcium nitrate and ammonium hydrogen phosphate with double-distilled water while stirring vigorously at room temperature. A dispersing agent solution, consisting of 0.2 vol.% of a 1:1 mixture of Darvan 821A and polyvinyl alcohol (PVA), was added to both solutions. The pH of each solution was adjusted to 10.5 using a 25% ammonium hydroxide (N$H_{4}$OH) solution.

The calcium nitrate solution was then combined with the ammonium hydrogen phosphate solution in accordance with the standard stoichiometry for pure hydroxyapatite (HAP), maintaining a calcium-to-phosphorus (Ca/P) ratio of 1.67. The synthesis occurred through the following reaction:$$10\mathrm{Ca}({NO}_{3}{)}_{2}\cdot 4H_{2}O\;+\;6({NH}_{4})2{HPO}_{4}+\;8{NH}_{4}\mathrm{OH}\;\to \;{Ca}_{10}({PO}_{4}{)}_{6}(\mathrm{OH}{)}_{2}+\;20NH_{4}{NO}_{3}+\;20H_{2}O$$

The (${NH}_{4}$)2${HPO}_{4}$ solution was added dropwise to the Ca$({NO}_{3}{)}_{2}$ solution at a temperature of 70 °C, ensuring continuous stirring and maintaining the pH at 10.5 with the 25% N$H_{4}$OH solution. The resulting suspension was stirred for 12 h. Afterward, the precipitate formed was filtered and washed three times with double-distilled water and anhydrous ethanol. Finally, the nHAP particles were lyophilized using a Christ alpha 1-4LD Plus model and dried in an oven at 80 °C for 12 h. After drying, the powder was crushed and subjected to heating at 300 °C in a furnace for 6 h to produce a fine dry powder of nHAP. The entire synthesis process has been described in another article [10].

#### 2.2.2. Production of Membranes

The membranes were produced using the electrospinning technique. A total of 2 g of polycaprolactone (PCL) was combined with 10% wt of nanohydroxyapatite (nHAP) and 2% wt of gentamicin sulfate (GEN). Both GEN and PCL were dissolved in a solvent mixture of chloroform and methanol at a ratio of 3:1, and the solution was stirred for 3 h at room temperature. After this, nHAP was added to the mixture, and stirring continued for an additional 9 h at the same temperature. The resulting composition was subjected to sonication in an ultrasonic bath at 40 kHz for 30 min before being loaded into an electrospinning syringe. The electrospinning process was carried out using an experimental setup at Babeș-Bolyai University—Raluca Ripan Chemical Research Institute in Cluj-Napoca. A voltage between 12 and 17 kV was applied, using a syringe needle with a diameter of 22G and a flow rate of 2.5 mL/h. The fibers were collected on a metallic surface covered with aluminum foil, positioned 23 cm away from the tip of the needle.

#### 2.2.3. Physicochemical Characterization

X-ray diffraction (XRD) measurements were conducted using a Shimadzu 6000 XRD diffractometer (Jasco International, Tokyo, Japan), equipped with a graphite monochromator for Cu-Kα radiation, at room temperature. Attenuated total reflectance Fourier transform infrared spectroscopy (FT-IR-ATR) was performed in the attenuated total reflectance mode. For spectrum collection, a FTIR spectrophotometer (FTIR-610, Jasco International Co., Ltd., Tokyo, Japan) was utilized, featuring an ATR attachment with a horizontal ZnSe crystal. The spectra were scanned in the mid-infrared range from 400 to 4000 cm^−1^. The spectral resolution was set to 4 cm^−1^, and each scan was repeated 100 times. The resulting spectra were corrected against the background spectrum.

#### 2.2.4. Structural Characterization

Transmission electron microscopy (TEM) investigations were conducted using a Hitachi H-7650 automatic microscope (Hitachi, Japan) operating at a voltage of 80 kV and a magnification of 20× to analyze the size and morphology of nHAP. The membrane’s structure was examined using scanning electron microscopy (SEM Inspect S, FEI, Eindhoven, The Netherlands) under high vacuum conditions at 15 kW, with a magnification of 1000× and a working distance of 10.7–13.9 mm. The diameter of nHAP from the TEM images and the fiber diameter from the SEM images were measured using ImageJ software (Version 1.54) (ANOVA) (National Institutes of Health, Bethesda, MD, USA).

## 3. Results

In this study, TEM (transmission electron microscopy) investigation (Figure 1) of crystals of the obtained nHAP particles exhibited a rod-like shape with an average size of approximately 26.39 nm (Figure 2). The results indicate that the synthesized particles fall within the “nano” range. Nanometer-sized nHAP particles are associated with low in vitro cytotoxicity, demonstrating good cell adhesion and promoting human osteoblast growth [19,20,21]. This has also been supported by other studies [22,23,24], which selected nHAP as an inorganic filler due to its bone regeneration properties. The XRD measurements were run at room temperature and the pattern of the obtained membrane (Figure 3) shows crystalline characteristics similar to those of the precursors, with diffraction peak positions approximately matching those of the precursors [25,26]. This result suggests that no new crystalline phase was formed in the membrane. The FTIR spectra of the various precursors (PCL, nHAP, and GEN) as well as the PCL matrix are presented in Figure 4. The FTIR spectra of GEN sulfate highlighted typical absorption bands at 1620, 1523, and 1288 cm^−1^, corresponding to the amide I, amide II, and amide III bonds of GEN, respectively. The peak observed at 1034 cm^−1^ was attributed to the HSO4^−1^ group, while the peak at 606 cm^−1^ corresponded to the SO2 band [13,19,27]. The PCL spectrum allowed for the identification of absorption bands at 2943 cm^−1^, associated with asymmetric CH2 stretching, and at 2866 cm^−1^, linked to symmetric CH2 stretching. A strong band at 1722 cm^−1^ corresponded to the carbonyl (C=O) stretching mode. Bands at 1240 cm^−1^ were identified as asymmetric C=O=C stretching, and bands up to 1169 cm^−1^ were attributed to symmetric C=O=C stretching [22,28].

The SEM image (Figure 5) displays randomly oriented bead-free fibers, building a porous structure with interconnected macropores resulting from the electrospinning process. The frequency distributions mean of fiber showed values between 2.18 and 15.75 µm. These findings are consistent with those reported in other studies [29,30]. The structure obtained through electrospinning is desirable as it mimics the extracellular matrix [31] creating an optimal microenvironment for cell growth, facilitating adhesion, proliferation, and differentiation, as evidenced by in vivo studies [11,32,33].

## 4. Discussion

When comparing the PCL-based membranes with Jason membranes, several key differences emerge in their respective characteristics and performance in guided bone regeneration (GBR):
Physicochemical Properties:
∘Mechanical Strength: PCL membranes typically exhibit superior mechanical stability compared to Jason membranes. This enhanced strength is beneficial for maintaining the structural integrity of the membrane during the healing process, particularly in complex or customized GBR applications. Conversely, while Jason membranes have adequate mechanical properties, they may not provide the same level of support in challenging scenarios.∘Resorption Rate: Jason membranes generally resorb more quickly than PCL membranes, which are designed for longer-term stability. This rapid resorption allows for quicker tissue integration but may limit their application in situations where prolonged support is necessary.
Biological Behavior:
∘Biocompatibility: The Jason membrane, being derived from natural collagen, is known for its high biocompatibility, promoting better integration with surrounding tissues. In contrast, while PCL is biocompatible, it is a synthetic polymer, and its natural integration may not match the rapidity seen with Jason membranes.∘Integration Speed: Jason membranes often integrate rapidly into the biological environment, facilitating quicker healing. This rapid integration can be advantageous in clinical scenarios requiring fast recovery. In contrast, PCL membranes, although slower to integrate, provide sustained mechanical support that can be crucial for long-term regeneration.
Durability and Stability:
∘Long-Term Performance: PCL membranes demonstrate durability over extended periods, making them particularly suitable for complex or customized GBR applications. This long-term stability is vital in situations where bone regeneration may take considerable time. In contrast, Jason membranes, while effective for certain applications, may not offer the same level of long-term support.
Clinical Applicability:
∘Indications for Use: The choice between PCL and Jason membranes may depend on the specific clinical scenario. For straightforward defects, Jason membranes may be preferred due to their rapid integration and biocompatibility. However, for more complex defects requiring custom solutions and extended support, PCL membranes may be the better choice.


A comparative analysis of polycaprolactone (PCL) membranes and Jason membranes reveals substantial differences in their physicochemical and biological characteristics, which directly influence their clinical applications in guided bone regeneration (GBR). From a mechanical standpoint, PCL membranes exhibit superior structural stability, an essential attribute for maintaining the regenerative space in complex defects or in cases requiring customized membrane designs. Although Jason membranes provide adequate mechanical support for routine clinical situations, their stability may be insufficient under challenging conditions where membrane deformation could compromise the regenerative outcome. The resorption rate represents another key differentiating factor. Owing to their natural collagen composition, Jason membranes undergo rapid biodegradation, which promotes fast tissue integration and accelerates healing. However, this rapid resorption can be a limitation in large or demanding defects where long-term structural support is required. In contrast, PCL membranes degrade slowly, offering sustained stability over extended periods and ensuring adequate space maintenance throughout the entire bone regeneration process.

From a biological perspective, Jason membranes excel in biocompatibility. Being derived from natural collagen, they are readily recognized by the host tissues, leading to rapid integration and a reduced inflammatory response. Although PCL is also biocompatible, its synthetic nature results in a slower integration rate. However, this gradual incorporation can be advantageous in situations where extended mechanical support is beneficial for long-term regenerative processes. These distinctions directly shape the clinical applicability of each membrane type. Jason membranes are particularly advantageous in straightforward or moderate defects where rapid healing and early tissue integration are desired. Conversely, PCL membranes are better suited for complex defects requiring customized solutions, prolonged stability, and predictable space maintenance. Ultimately, the optimal membrane selection relies on a careful assessment of the regenerative requirements of each clinical case to ensure predictable and successful therapeutic outcomes.

Overall, while both PCL and Jason membranes have their respective advantages, the decision on which membrane to use in GBR procedures should be guided by the specific clinical context, desired outcomes, and the nature of the bone defect. Jason membranes, with their rapid resorption and high biocompatibility, are ideal for straightforward defects requiring fast integration and early healing. In contrast, PCL membranes provide superior mechanical stability and long-term durability, making them more appropriate for complex or customized defects where extended support is critical. Selecting the optimal membrane should therefore be guided by the specific clinical requirements, balancing the need for rapid biological integration with sustained structural support to achieve predictable and effective regenerative outcomes.

## 5. Conclusions

This study presents a thorough investigation into the use of polycaprolactone (PCL) and Jason^®^ membranes in the context of guided bone regeneration (GBR). The comparative analysis highlights the distinct physicochemical properties, biological behaviors, and clinical applicability of these two types of membranes, emphasizing their unique strengths and potential use cases in regenerative medicine. Based on these findings, our innovative electrospun PCL membrane using the synthesized nanohydroxyapatite (nHAP) demonstrated favorable characteristics, including small particle size and rod-like morphology, which are advantageous for enhancing cellular adhesion and proliferation. These properties contribute to the effectiveness of bone regeneration, supporting the notion that nanometer-sized nHAP can minimize cytotoxicity while promoting osteoblast activity. The successful integration of nHAP within PCL-based membranes holds promise for improved mechanical stability and durability, making them suitable for complex clinical scenarios where long-term support is essential.

The electrospinning technique employed for membrane production yielded a fibrous structure that closely mimics the extracellular matrix, further facilitating cell growth and differentiation. The physicochemical characterization through X-ray diffraction and Fourier transform infrared spectroscopy confirmed the integrity and functional properties of the membranes, while electron microscopy techniques provided insights into their structural features.

This study emphasizes the critical role of membrane selection in GBR procedures. Jason^®^ membranes offer rapid integration and high biocompatibility for straightforward cases, while PCL-based membranes provide enhanced mechanical stability and long-term support for complex defects. Future research should explore material optimization and potential synergistic applications to further improve clinical outcomes in bone regeneration.

## Figures and Tables

**Figure 1 bioengineering-13-00023-f001:**
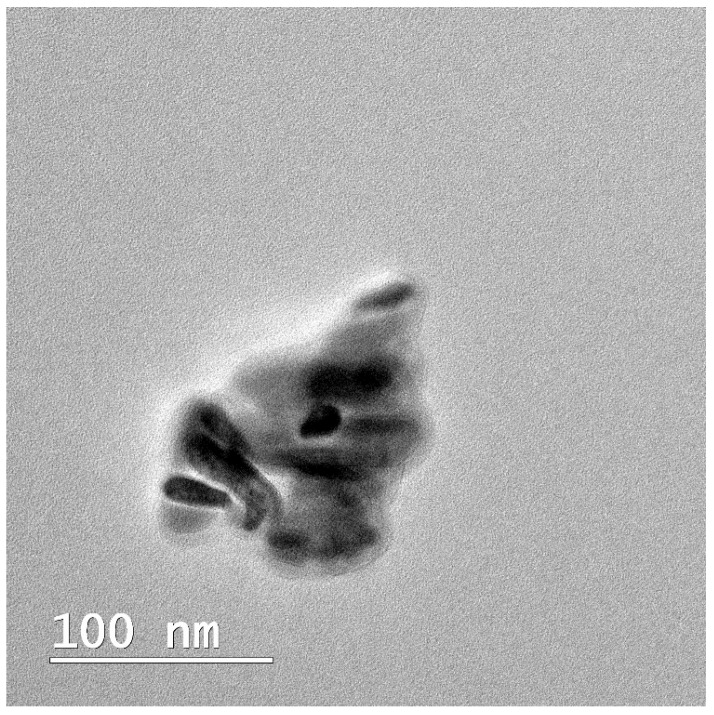
TEM image of the obtained nHAP.

**Figure 2 bioengineering-13-00023-f002:**
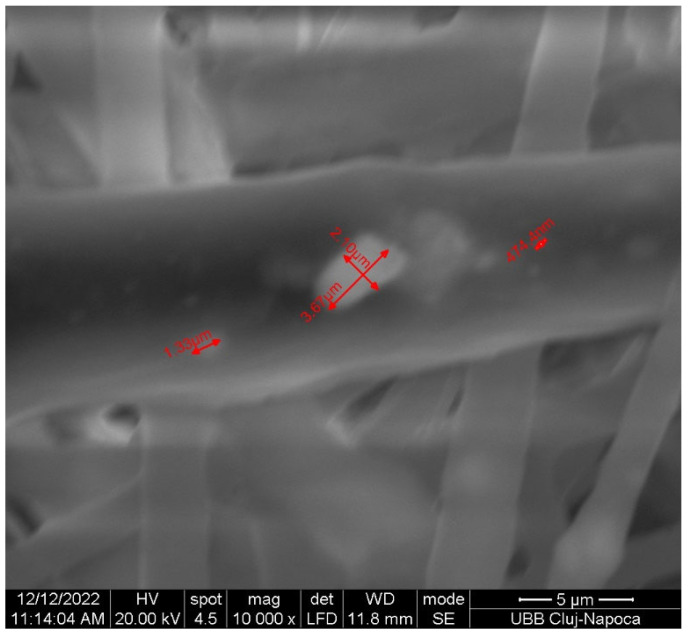
nHAP particle size in the fiber.

**Figure 3 bioengineering-13-00023-f003:**
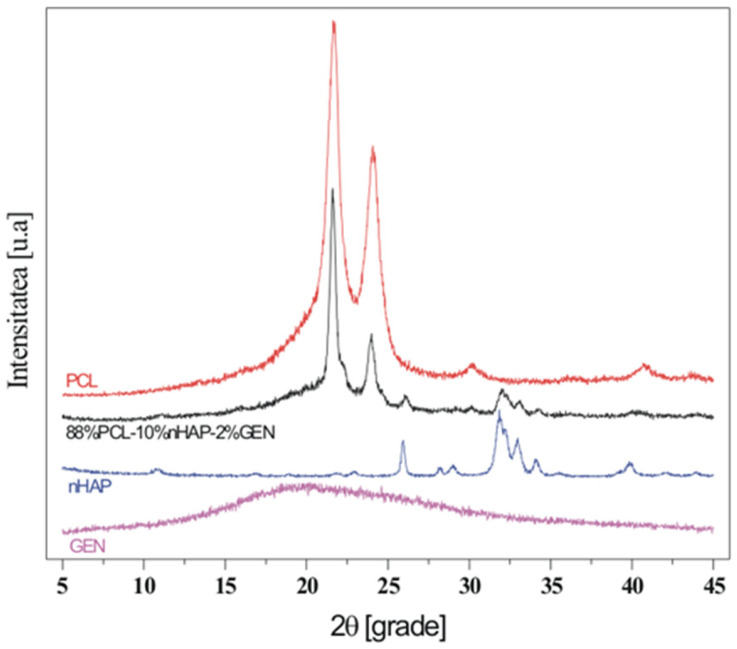
XDR image of the fabricated membrane.

**Figure 4 bioengineering-13-00023-f004:**
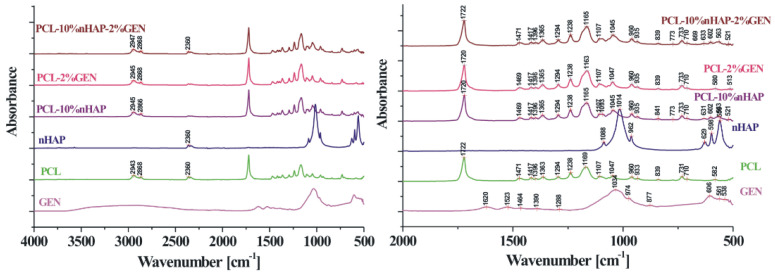
FTIR image of the fabricated membrane.

**Figure 5 bioengineering-13-00023-f005:**
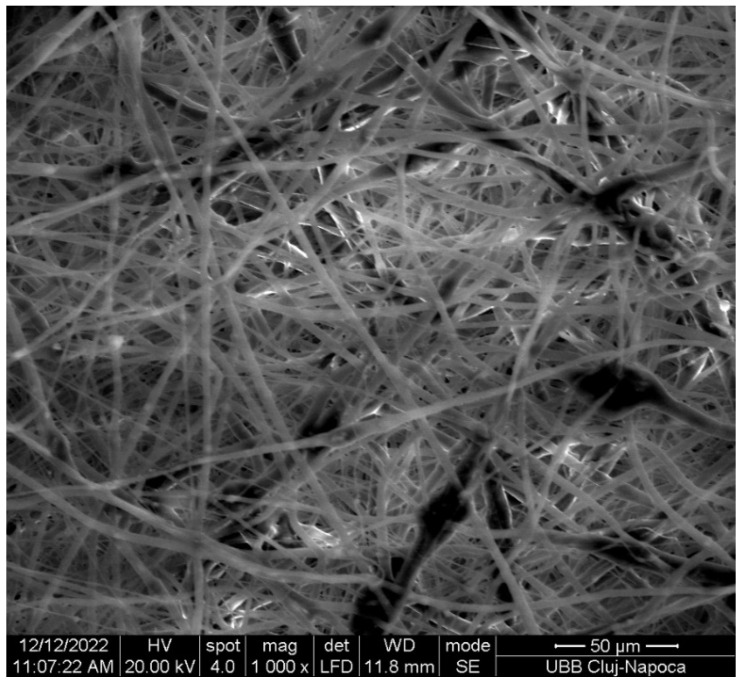
SEM image of the fabricated membrane.

**Table 1 bioengineering-13-00023-t001:** Physicochemical Properties.

Characteristic	Jason Membrane	PCL Membrane
Origin	Natural collagen types I and III (porcine pericardium)	Synthetic aliphatic polyester
Structure	Multilayered, with dense collagen fibers	Semicrystalline, with controllable porosity
Degradation Method	Enzymatic (collagenases)	Hydrolysis (esters → caproic acid)
Resorption Time	3–6 months	6–24 months (depending on design and thickness)
Degradation Products	Physiological amino acids	Fatty acids with potential local irritative effects

**Table 2 bioengineering-13-00023-t002:** Comparative Analysis of Mechanical Properties.

Parameter	Jason	PCL
Tensile Strength	High for collagen	Superior, supports volumetric stability
Flexibility	High, allows for easy adaptation	Rigid/semi-rigid, requiring pre-shaping
Manipulability	Easy to handle, non-sticky	Requires special handling or thermal pre-forming
Fixation	Pins, screws, or sutures	Often requires specific screws or pins

**Table 3 bioengineering-13-00023-t003:** Biological Integration and Cost Analysis.

Parameter	Jason	PCL
Biological Integration	Very rapid	Slow
Dimensional Stability	Limited (3–6 months)	Excellent (6–24 months)
Cost	Moderate	High (especially for 3D printed variations)
Customization	No	Yes—CAD/CAM, 3D printing

**Table 4 bioengineering-13-00023-t004:** Clinical Applicability.

Clinical Situation	Jason	PCL
Socket Preservation	Ideal	Rarely used
Sinus Lift	Frequently utilized	Used in customized variations (e.g., 3D print)
Horizontal Augmentation	Suitable for moderate defects	Recommended for long-term stability
Vertical Augmentation	Limited	Indicated (due to rigidity)
Peri-implant Defects	Yes	Yes, in customized forms

## Data Availability

The original contributions presented in the study are included in the article, further inquiries can be directed to the corresponding author.

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
