# Peer review of "Integrated Analysis of an Innovative Composite Polycaprolactone Membrane and a Jason Membrane in Guided Bone Regeneration"

_bioengineering, 2025, doi:10.3390/bioengineering13010023_

Round 1

Reviewer 1 Report

Comments and Suggestions for Authors

 Integrated analysis of an innovative composite polycaprolactone tone membrane and Jason membrane in guided bone regeneration

Abstract and introduction

  • The text and abbreviations imply gentamicin sulfate (GEN) in one place and genipin (GEN) in another; FTIR assignments later align with gentamicin, while the membrane chemistry text reads like genipin crosslinking. This conflict begins in the opening framing. Gentamicin (antibiotic) and genipin (crosslinker) have opposite functions, spectra, and regulatory implications. The central claim (antimicrobial vs crosslinked mechanics) depends on this.
  • The Introduction lists multiple advantages and clinical scenarios. Still, it does not specify the target defect type (e.g., horizontal augmentation, vertical augmentation, peri-implant defects) for which the new membrane is primarily intended. Design requirements (thickness, stiffness, degradation time, porosity, antibiotic load) strongly depend on defect type and anatomical site.
  • It is recommended to use the following paper. 4D printing‐encapsulated polycaprolactone–thermoplastic polyurethane with high shape memory performances.

Materials and methods

  • Materials list “gentamicin sulfate (GEN)”, while the membrane recipe uses “2 wt% genipin (GEN)”. FTIR assignments are then interpreted for GEN sulfate (amide bands, HSO4−), consistent with gentamicin, not genipin. Gentamicin (an aminoglycoside antibiotic) and genipin (a natural crosslinker) have opposite functions, spectra, release/toxicity profiles, and regulatory implications. All downstream claims (antimicrobial vs crosslinking) hinge on this identity.
  • Correct XDR → XRD; harmonize units (nm/µm) and symbols; avoid mixing “um” and “µm.”
  • Did you perform any measurement of porosity (e.g., image analysis, mercury porosimetry) or only qualitative SEM observation? If only qualitative, this should be acknowledged as a limitation

Results and discussion

  • Results state nHAP has ~26.39 nm average size, but the very following sentence says “length 0.474–1.33 µm, width ~2.10 µm” for “the nHAP fibers,” which would make the width larger than the length and conflicts with the “nano” claim. Later, the electrospun fiber diameter is reported as 15.2 µm, which is atypically large for electrospinning and is not reconciled with prior values. Additionally, TEM is described as “operating at 120 kV, at a voltage of 80 kV”—two incompatible settings. Size, scale bars, and instrument settings underpin all structure–property claims. Conflicting magnitudes and typos (XDR) indicate potential data/caption mismatches.
  • The text claims advantages related to low cytotoxicity, osteoblast adhesion, proliferation, and bone regeneration. Still, the Methods include no in vitro tests (cell culture, viability, ALP, mineralization) and no antibacterial tests (MIC, inhibition zones, planktonic or biofilm models). No biological or antibacterial assays are described. Without these methods, the study cannot directly support claims about GBR performance, infection control, or long-term clinical suitability
  • There is no experimental comparison with Jason membranes or other commercial GBR membranes, despite extensive comparative discussion elsewhere in the paper.

Author Response

I made all the requested changes, rephrased a few sentences, and added bibliographical sources to support the ideas in the article. Thank you so much for your time

Reviewer 2 Report

Comments and Suggestions for Authors

This paper describes the preparation of a novel electrospinning polycaprolactone (PCL) membrane and the addition of novel synthetic nano-hydroxyapatite to enhance its specific beneficial properties for targeted reconstruction. The article is well-organized, but there are still some problems:

  1. The introduction part is suggested to add relevant progress, highlight the innovation of this paper, and be cited and discussed.
  2. It is recommended to present lines 48-52 in a table format for better standardization.
  3. The chemical formulas in the paper are not standardized, such as the number of 70-71 lines should be unified.
  4. The paper recommends harmonizing the figure sizes. Figures 3 and 4 should be aligned with consistent font sizes and positioned appropriately. Since TEM and XRD experimental results are not presented separately, each figure could be described with its function and analyzed. Note that the spacing between PCL-10%nHAP and nHAP lines in Figure 4 is problematic, as they already intersect; adjustments are advised.
  5. Some parts of the discussion are recommended to be described in paragraph form.
  6. Conclusion Some of the conclusions are suggested to be concise and highlight the important conclusions.

7 The formatting of table annotations in the paper is incorrect.

8 Some sentences in the text lack conciseness, such as lines 183-188. While comparing PCL membranes with Jason membranes demonstrates that PCL membranes exhibit superior mechanical strength, the phrasing requires refinement.

9 Lines 205-216 in the conclusion section use excessive “possibly” statements. Sentences should be more precise where appropriate.

10 Raw material descriptions require optimization. When materials are first introduced, include parenthetical notes specifying composition, origin, etc.

Author Response

  1. The introduction part is suggested to add relevant progress, highlight the innovation of this paper, and be cited and discussed.

Response: 

Bone regeneration is a complex and tightly orchestrated sequence of physiological events that occurs continuously throughout life, as bone tissue undergoes constant repair and remodeling. Conventional reconstruction approaches, such as autografts and allografts, present significant limitations in addressing complex defects and restoring the native structural and functional architecture. In this context, tissue engineering offers a promising therapeutic strategy focused on regenerating rather than merely replacing damaged tissue.

Guided bone regeneration (GBR) relies on the use of resorbable or non-resorbable barriers to exclude soft tissues from the bone regeneration site. Within bone tissue engineering (BTE), biomaterials serve as artificial extracellular matrices that support cell adhesion, proliferation, and differentiation, thereby promoting the formation of newly developed bone tissue(1,2). The complex interplay among proteins, proteoglycans, angiogenic growth factors, and matrix-associated vesicles within the extracellular matrix (ECM) directs host cells toward regenerative behaviors, promoting tissue-specific differentiation and injury resolution (3). However, advanced fabrication of ECM-based materials remains challenging, and current clinical applications of ECM scaffolds are largely confined to native tissue forms or processed powders. A broad range of collagen-based scaffolds has been developed for various regenerative applications, with these structures being designed to elicit favorable biological responses and to mimic the properties of native extracellular matrices(4).

This work provides a critical assessment of the hierarchical structure and key properties of native collagen, with particular reference to the Jason MegaGen membrane, and examines the current challenges associated with the fabrication of polycaprolactone-based materials. Specific attention is given to the need for tailoring their physicochemical and mechanical properties, as well as their geometrical configurations, to meet the requirements of targeted biomedical applications, with a special emphasis on advanced solutions for bone tissue engineering(5–7). Jason® Membrane (MegaGen), are widely utilized due to their rapid biological integration; however, in recent years, synthetic membranes made of polycaprolactone (PCL) have gained popularity due to their remarkable physicomechanical properties (8,9).

Research indicates that PCL membranes not only provide superior mechanical strength but also enhance cellular proliferation, contributing to more effective bone regeneration outcomes. The ongoing advancements in GBR materials underscore the necessity for continued investigation in this field to optimize clinical results and improve patient quality of life. 

The objective and innovative aspect of this research involved the synthesis of novel nanohydroxyapatite (nHAP) using two different dispersing agents, as well as the application of this new nHAP in developing a new electrospun membrane (EM) based on polycaprolactone (PCL) (9). The newly synthesized nHAP was characterized using transmission electron microscopy (TEM), X-ray diffraction (XRD), and attenuated total reflectance Fourier transform infrared spectroscopy (FTIR-ATR). Additionally, all precursors and electrospun membranes were structurally characterized through XRD and FTIR-ATR analyses. For mechanical characterization, parameters such as maximum load force, Young's modulus, stiffness, and tensile strength of the electrospun membranes were evaluated. Scanning electron microscopy (SEM) was employed to investigate the topography of the membranes.

2. It is recommended to present lines 48-52 in a table format for better standardization.

Response: I formatted the information and transposed it into a table as found in the attached document.

3.The chemical formulas in the paper are not standardized, such as the number of 70-71 lines should be unified.

Response: I formatted the information as found in the attached document.

4.The paper recommends harmonizing the figure sizes. Figures 3 and 4 should be aligned with consistent font sizes and positioned appropriately. Since TEM and XRD experimental results are not presented separately, each figure could be described with its function and analyzed. Note that the spacing between PCL-10%nHAP and nHAP lines in Figure 4 is problematic, as they already intersect; adjustments are advised.

5.Some parts of the discussion are recommended to be described in paragraph form.

Response: 

A comparative analysis of polycaprolactone (PCL) membranes and Jason membranes reveals substantial differences in their physicochemical and biological characteristics, which directly influence their clinical applications in guided bone regeneration (GBR). From a mechanical standpoint, PCL membranes exhibit superior structural stability, an essential attribute for maintaining the regenerative space in complex defects or in cases requiring customized membrane designs. Although Jason membranes provide adequate mechanical support for routine clinical situations, their stability may be insufficient under challenging conditions where membrane deformation could compromise the regenerative outcome. The resorption rate represents another key differentiating factor. Owing to their natural collagen composition, Jason membranes undergo rapid biodegradation, which promotes fast tissue integration and accelerates healing. However, this rapid resorption can be a limitation in large or demanding defects where long-term structural support is required. In contrast, PCL membranes degrade slowly, offering sustained stability over extended periods and ensuring adequate space maintenance throughout the entire bone regeneration process.

From a biological perspective, Jason membranes excel in biocompatibility. Being derived from natural collagen, they are readily recognized by the host tissues, leading to rapid integration and a reduced inflammatory response. Although PCL is also biocompatible, its synthetic nature results in a slower integration rate. However, this gradual incorporation can be advantageous in situations where extended mechanical support is beneficial for long-term regenerative processes. These distinctions directly shape the clinical applicability of each membrane type. Jason membranes are particularly advantageous in straightforward or moderate defects where rapid healing and early tissue integration are desired. Conversely, PCL membranes are better suited for complex defects requiring customized solutions, prolonged stability, and predictable space maintenance. Ultimately, the optimal membrane selection relies on a careful assessment of the regenerative requirements of each clinical case to ensure predictable and successful therapeutic outcomes.

Overall, while both PCL and Jason membranes have their respective advantages, the decision on which membrane to use in GBR procedures should be guided by the specific clinical context, desired outcomes, and the nature of the bone defect. Jason membranes, with their rapid resorption and high biocompatibility, are ideal for straightforward defects requiring fast integration and early healing. In contrast, PCL membranes provide superior mechanical stability and long-term durability, making them more appropriate for complex or customized defects where extended support is critical. Selecting the optimal membrane should therefore be guided by the specific clinical requirements, balancing the need for rapid biological integration with sustained structural support to achieve predictable and effective regenerative outcomes.

6,7 ,8, 9 ,10.Conclusion Some of the conclusions are suggested to be concise and highlight the important conclusions.

Response: I made the changes and you can find them in the attached document.

Round 2

Reviewer 1 Report

Comments and Suggestions for Authors

Accept as is.

Author Response

Your recommendations were well received and really helpful. Thank you very much for your time and effort.

Reviewer 2 Report

Comments and Suggestions for Authors

The manuscript does not provide point-by-point responses to the reviewers' questions.
The tables and formatting in the manuscript are not well-structured.
I recommend rejection for resubmission.

Author Response

I have responded to each recommendation individually(see attachment). I have also revised and formatted the tables so that they are according to the journal's standards, easy to read and interpret. Otherwise, please be a little more explicit when you say that they are not well structured. Thank you very much for your time.

Round 3

Reviewer 2 Report

Comments and Suggestions for Authors

 The research topic is relevant, but the manuscript contains certain formatting errors and minor content-related issues that need improvement. My specific suggestions are as follows:

  1. The abstract section of the article states that a novel proprietary reinforcement membrane was developed through comprehensive comparison of membranes. It directly introduces Jason® membrane and PCL membrane for comparison. Although both membranes exhibit superior performance, the rationale for selecting these two membranes is not clearly explained.
  2. The text begins with a list-style comparison starting from line 85, but the conclusions drawn afterward lack clarity or sufficient logical connection.
  3. Adjust the image proportions and standardize the numeric format in the original file.
  4. The article's headings and numbering are not standardized, and their meanings are unclear.
  5. The citation marks in the article are not standardized.

Author Response

  1. The abstract section of the article states that a novel proprietary reinforcement membrane was developed through comprehensive comparison of membranes. It directly introduces Jason® membrane and PCL membrane for comparison. Although both membranes exhibit superior performance, the rationale for selecting these two membranes is not clearly explained.

In the context of guided bone regeneration (GBR), the selection of an appropriate resorbable membrane plays a crucial role in the clinical success of the procedure. Precise knowledge about the distinct differences in properties is fundamental for correct selection of the membrane. This article presents an integrated comparative analysis between membranes, conducted for this given purpose and one step beyond: to fabricate a novel membrane with dedicated enhanced properties according to the targeted function. Our previous analysis showed that polymer membranes that met most histopathological criteria also produced the most remarkable results when radiologically observed. The top effective scaffolds were those containing active macromolecules released conditionally and staged. The PLGA and polycaprolactone scaffolds were found in this category and they granted a marked increase in bone density and improvement in osteoinduction(1). Based on these results, we decided to create a new polycaprolactone membrane in order to compare it with a standard currently on the market, Jason membrane.

2. The text begins with a list-style comparison starting from line 85, but the conclusions drawn afterward lack clarity or sufficient logical connection.

I have reformulated the relevant conclusions and added them to the final article.

3. Adjust the image proportions and standardize the numeric format in the original file.

I adjusted the size and renumbered the images in the original article.

4.The article's headings and numbering are not standardized, and their meanings are unclear.

I made the changes to the original article, I hope they are now clearer and easier to follow.

5.The citation marks in the article are not standardized.

I put them all in Vancouver style.
